# Paravertebral Block Versus Preemptive Ketamine Effect on Pain Intensity after Posterolateral Thoracotomies: A Randomized Controlled Trial

**DOI:** 10.3390/jcm9030793

**Published:** 2020-03-14

**Authors:** Michał Borys, Agata Hanych, Mirosław Czuczwar

**Affiliations:** 1Second Department of Anesthesia and Intensive Therapy, Medical University of Lublin, 20-059 Lublin, Poland; miroslaw.czuczwar@umlub.pl; 2Department of Anesthesia and Intensive Therapy, Podkarpackie Center of Lung Disease, 35-241 Rzeszów, Poland; agahanych@gmail.com; 3Department of Anesthesia, Intensive Therapy and Pain Treatment, 39-120 Sędziszów Małopolski, Poland

**Keywords:** ketamine, paravertebral block, osterolateral thoracotomy, thoracotomy, visual analog scale

## Abstract

Severe postoperative pain affects most patients after thoracotomy and is a risk factor for post-thoracotomy pain syndrome (PTPS). This randomized controlled trial compared preemptively administered ketamine versus continuous paravertebral block (PVB) versus control in patients undergoing posterolateral thoracotomy. The primary outcome was acute pain intensity on the visual analog scale (VAS) on the first postoperative day. Secondary outcomes included morphine consumption, patient satisfaction, and PTPS assessment with Neuropathic Pain Syndrome Inventory (NPSI). Acute pain intensity was significantly lower with PVB compared to other groups at four out of six time points. Patients in the PVB group used significantly less morphine via a patient-controlled analgesia pump than participants in other groups. Moreover, patients were more satisfied with postoperative pain management after PVB. PVB, but not ketamine, decreased PTPS intensity at 1, 3, and 6 months after posterolateral thoracotomy. Acute pain intensity at hour 8 and PTPS intensity at month 3 correlated positively with PTPS at month 6. Bodyweight was negatively associated with chronic pain at month 6. Thus, PVB but not preemptively administered ketamine decreases both acute and chronic pain intensity following posterolateral thoracotomies.

## 1. Introduction

Even with recent advances in pain management, moderate to severe postoperative pain still affects many patients after open-chest surgeries or thoracotomies [1]. Factors contributing to the development of postoperative pain include skin incision, damage to ribs and muscles, intercostal nerve injury, and pleural irritation [2]. Moreover, severe pain is a risk factor for post-thoracotomy pain syndrome (PTPS). PTPS affects 50% of patients at six months after surgery [3].

The combination of systemic analgesics with regional anesthesia techniques is the most preferred pain management approach in patients undergoing thoracotomies [4]. Among the different regional analgesia techniques, thoracic epidural analgesia (TEA) is most commonly performed by anesthesiologists [5,6]. Paravertebral block (PVB) appears to be as effective as TEA for postoperative pain control after lung surgery [7]. Although regional anesthesia techniques are advantageous to systemic analgesia during the perioperative period after thoracic surgery, alternative approaches are used because of failed blocks, lack of patient consent, and contraindications for this procedure [8,9].

Ketamine is a general anesthetic with analgesic properties. Its action is mediated via N-methyl-D-aspartate (NMDA)-receptors [10]. Many recent studies endorse the analgesic features of ketamine [11,12,13,14], but only a few trials directly compare ketamine with regional anesthesia techniques in patients undergoing thoracic surgery [15,16,17]. 

This study aimed to compare pain intensity in patients after preemptive ketamine versus PVB during scheduled posterolateral thoracotomy, with further analysis of morphine consumption, patient satisfaction, and chronic pain evaluation.

## 2. Materials and Methods

### 2.1. Ethical Considerations

The randomized, controlled trial was conducted in a thoracic surgery department of a teaching hospital. The study protocol was approved by the Institutional Review Board of the Medical University of Lublin, Lublin, Poland (permit number KE-0254/84/2016, 24 March 2016, and registered with the Australian New Zealand Clinical Trial Registry (ACTRN12616000900415) before patient recruitment. Written informed consent was obtained from each patient, and the study was conducted in accordance with the tenets of the Declaration of Helsinki for medical research involving human subjects.

### 2.2. Patient Selection

The inclusion criteria identified patients scheduled for posterolateral thoracotomy and those whose age was >18 and <75 years. Patients with known coagulopathy, poorly controlled diabetes mellitus, depression or other psychiatric disorders that required antidepressant drugs, alcohol or recreational drug addiction, or an allergy to the studied drugs were excluded. Before study inclusion, each patient was tested for the capability to understand the visual analog scale (VAS), operate a patient-controlled analgesia (PCA) pump, answer a phone call, and comprehend the Neuropathic Pain Syndrome Inventory (NPSI). Patients who did not fulfill these criteria were excluded from the study. Individuals older than 75 years were presumed to have difficulties with answering phone calls and describing chronic pain on the NPSI.

### 2.3. Study Groups and Intervention

The patients were randomly allocated to one of three groups (detailed below) by computer-generated randomization conducted by a team member who was not involved in patient assessment or surgery. The same team member prepared opaque envelopes in which the intervention type was concealed. These envelopes were opened when the patient was present in the operating theater.

Each patient was anesthetized in the same manner. The drugs used for induction of general anesthesia were propofol (1–2 mg/kg, Propofol 1% Fresenius, Fresenius Kabi Deutschland GmbH, Bad Homburg, Germany), fentanyl (1–3 µg/kg, Fentanyl WZF, Warszawskie Zakłady Farmaceutyczne Polfa S.A., Warszawa, Poland), and rocuronium (0.6–1.0 mg/kg, Rocuronium Kabi, Fresenius Kabi Polska Sp. z o.o., Warszawa, Poland). Suxamethonium was used (1.0 mg/kg, Chlorsuccillin, PharmaSwiss Česká Republika s.r.o., Prague, Czech Republic) if required. A double-lumen endotracheal tube was inserted, and the ipsilateral lung was deflated during the procedure. Maintenance was provided with continuous infusion of propofol (4–8 mg/kg/h) and additional boluses of fentanyl and rocuronium.

In the PVB group, an ipsilateral block was performed by the same anesthesiologist before the induction of general anesthesia. The loss of resistance technique was used for identifying the PVB space and catheter (Perifix 701, B. Braun, Melsungen, Germany) placement. The solution of bupivacaine (30 mL, 0.25%, Bupivacainum Hydrochloricum WZF, Warszawskie Zakłady Farmaceutyczne Polfa S.A., Warszawa, Poland) with fentanyl (2 µg/mL) was deposited after proper identification of the PVB space.

In the ketamine group (KET), after the induction of general anesthesia but before the first incision, patients received a bolus of ketamine (1 mg/kg, Ketalar, Pfizer Europe MA EEIG, Bruxelles, Belgium) intravenously. Patients in the control group (CON) did not receive any additional intervention.

Approximately 30 min before the end of the surgery, each patient received morphine (0.15 mg/kg, Morphini sulfas WZF, Warszawa, Poland) and metamizole (2.0 g, Pyralgin, Zakłady Farmaceutyczne Polpharma, Starogard Gdański, Poland) intravenously.

### 2.4. Postoperative Pain Control

Each patient was admitted to the postoperative care unit where vital signs—respiratory rate, plethysmography, oxygen saturation, heart rate, and blood pressure—were monitored. Postoperative pain was treated with morphine and metamizole. Morphine was administered via the PCA pump (bolus 1.0 mg, lockout period 5 min Perfusor space PCA, B.Braun, Melsungen, Germany). In case of severe pain or ≥40 mm on the VAS, up to two doses of morphine (5 mg) were administered based on the nurse’s discretion. In the PVB group, the local anesthetic (5–8 mL/h) solution was continued in the postoperative period.

### 2.5. Outcomes

The primary outcome of the study was to evaluate pain severity on the VAS scale (0–100 mm) in patients after three types of analgesic approaches. Pain severity was assessed after patient transfer to the postoperative care unit and at hours 2, 4, 8, 12, and 24 after the surgery completion by nurses not directly involved in the study (blinded). 

The secondary outcomes included total morphine consumption via PCA pump during the first postoperative day, and patient satisfaction with pain management was assessed at the time of discharge from the hospital. Patients could describe their satisfaction with pain management as perfect (5), good (4), moderate (3), poor (2), or very poor (1). Moreover, PTPS was evaluated via a telephone interview after 1, 3 and 6 months from patients’ discharge. For chronic pain analysis, the NPSI by Bouhassira et al. was utilized [18]. This inventory was used in our previous study to detect persistent postsurgical pain [19].

### 2.6. Statistical Analysis

Normal distribution was checked for all continuous variables with the Shapiro–Wilk test. The analysis of variance (ANOVA) was used for normally distributed parameters. Normally distributed variables were presented as means (95% confidence intervals). The Kruskal–Wallis test by ranks was used to compare parameters with non-normal distribution. If the results of the Kruskal–Wallis test showed statistical significance, a pairwise comparison was performed with the Mann–Whitney U test and Bonferroni correction was applied. Medians (interquartile ranges) were used to present these data. Qualitative variables were compared with Fisher’s exact test. Logistic regression was applied to detect parameters affecting chronic pain presentation, and the odds ratio (OR) was used to describe PTPS predictors included in the model. The receiver operating characteristic (ROC) curve was calculated for the best model. All measurements were performed using Statistica 13.1 software (Stat Soft. Inc., Tulsa, OK, USA).

### 2.7. Power Analysis

The sample size analysis was calculated for the primary outcome of the current study. A preliminary study was performed to assess the sample size. Twenty patients (CON group = 10, PVB group = 10) were included. Mean VAS results for the CON and PVB groups were 59 and 40, respectively. Thus, 40 patients were to be recruited, 20 in each group, for significance level (α) = 0.05 and power = 0.8. Because three groups were used in the study, we decided to randomize 120 patients. 

## 3. Results

This single-center study was conducted in a thoracic surgery department of a teaching hospital between July 2016 and November 2018. Patient demographics are presented in Table 1. The detailed description of the screening, randomization, and follow-up process is presented in Figure 1.

### 3.1. Primary Outcome

The pain intensity measured by VAS was significantly different between the PVB group and other groups at four out of six time points, while no difference was found between the KET and CON groups. The detailed description of VAS measurements is presented in Table 2.

### 3.2. Secondary Outcomes

Morphine consumption: Patients in the PVB group (13 (7–21)) had significantly less morphine consumption via PCA than individuals in the KET (34.5 (15–49), *p* = 0.004) and CON (25 (24–39), *p* = 0.0001) groups. Moreover, patients required significantly less rescue doses of morphine in the postoperative period after continuous PVB (0 (0–0)) compared to participants in the KET (5 (0–5), *p* = 0.0004) or CON (0 (0–8), *p* = 0.001) groups. No difference in postoperative morphine consumption was noticed between the KET and CON groups either via PCA or rescue boluses.

Patient satisfaction: Patients in the PVB group were more satisfied with their pain management than individuals in other groups. The results of satisfaction presented as medians (interquartile ranges) for patients in PVB, KET, and CON groups were 4 (4–5), 4 (4–4) (*p* < 0.001) and 4 (3–4) (*p* < 0.001), subsequently.

Chronic postsurgical pain: PTPS severity was significantly lower in the PVB group compared to the KET and CON groups after 1, 3, and 6 months from hospital discharge. No difference in PTPS severity was noticed between KET and CON groups (Table 3). The number of patients who perceived PTPS did not vary between groups at months 1 and 3. However, significantly fewer patients in the PVB group had PTPS at month 6 (Table 4).

Acute and chronic pain association with the type of surgery: Seven types of surgical procedures were performed in the study including wedge resection (46), lobectomy (28), decortication (20), pneumonectomy (7), segmentectomy (6), tumor resection, and exploratory thoracotomy (2). Because only four patients had tumor resections and exploratory thoracotomies, they were excluded from statistical analysis. Only a single pairwise comparison showed a significant result. Pain intensity was significantly higher after lung decortication than segmentectomy after patients’ admission to the postoperative unit (0 h) and after two postoperative hours (2 h). The median pain severity was 73.5 (58–86) and 46.5 (42–56) after decortication, and 44 (22–51) and 23.5 (16–28) after segmentectomy. Bonferroni correction was applied for multiple pairwise comparisons and probability was set at 0.0051 to present significant results. No difference was found between the surgery type for postoperative morphine consumption. Moreover, the surgery type did not affect PTPS intensity in our study.

Post-thoracotomy pain syndrome prediction: Logistic regression analysis detected three variables to predict PTPS occurrence during the sixth month after hospital discharge. Acute pain intensity at hour 8 and chronic pain intensity at month 3 had a positive correlation with PTPS presence during month 6. The OR and 95% CI for acute pain at hour 8 and PTPS intensity at month 3 were 1.085 (1.025–1.148) and 1.182 (1.064–1.312), respectively. In contrast, patients’ weight had a negative association with PTPS occurrence with OR = 0.962 (0.939–0.986). The area under the ROC curve was 0.889 for this predictive model including three variables.

## 4. Discussion

The results of this study confirm the effectiveness of PVB in alleviating acute postoperative pain in patients undergoing posterolateral thoracotomy. Preoperative PVB lowers pain intensity on VAS and hence the patients used less morphine via PCA and were more satisfied with their pain management during the perioperative period. However, acute postoperative pain results were not different between patients treated with ketamine or no ketamine. The surgery type following thoracotomy had some impact on acute but not chronic pain severity. Patients experienced higher pain severity on VAS during the first two postsurgical hours after decortication procedures compared to lung segmentectomies.

In this study, most patients perceived PTPS at months 1, 3 and 6 after hospital discharge. Patients who received PVB experienced less PTPS intensity compared to controls and ketamine administration. Thus, preemptive ketamine did not alleviate PTPS severity and was comparable to the control group. Furthermore, the surgery type did not affect PTPS intensity. Acute pain intensity at hour 8, PTPS intensity at month 3, and low body weight of the patient influenced chronic pain occurrence during month 6.

In a meta-analysis of 14 studies, Yeung et al. reported equal efficacy of PVB and TEA in postoperative pain alleviation [7]. Moreover, minor complications such as hypotension, nausea and vomiting, pruritis, and urinary retention occurred less frequently after PVB. In our previous study, during the first postoperative hours, patients after posterolateral thoracotomies perceived severe pain which was comparable to upper abdominal surgery [20]. However, with continuous PVB, the pain intensity reduced drastically at postoperative hour 12 and was comparable in severity to the less painful procedures of the head and neck or lower abdominal surgeries.

In a systematic review of ketamine effect on acute and chronic postsurgical pain after thoracotomies by Moyse et al. [21], most studies used intravenous ketamine as a bolus or infusion [16,17,22,23,24,25,26,27], while others used intramuscular or epidural ketamine [16,28,29,30]. A standard regime for ketamine administration is not available. Some authors used ketamine while others used the dextrorotatory enantiomer, S (+)-ketamine. In most studies, the effect of ketamine on acute postoperative pain was only moderate, based on pain intensity measured by VAS or a numerical rating scale (NRS) [16,22,28,29]. Three studies showed a decrease in opioid consumption with ketamine [25,30,31], of which one study used PVB as a pain treatment strategy [25]. 

None of the seven studies showed a beneficial effect of preemptive ketamine administration on PTPS, even though some studies used NPSI to evaluate chronic pain intensity similar to this study [16,22,27]. Duale et al. reported a positive NPSI score in 58% of patients in the fourth month after the surgery. Tena et al. reported similar results at three months after hospital discharge, and approximately 30% of patients continued to experience PTPS during the sixth month. In this study, 63.1% of all patients had signs of persistent pain in the sixth month (Table 4), and pain severity was notably lower in patients after PVB (48%) by NPSI.

In the current study, PVB decreased PTPS intensity more significantly than ketamine. Weinstein et al. compared regional anesthesia with conventional treatment to study the incidence of persistent pain in a meta-analysis [32]. Interestingly, only a single study presented significant differences between epidural anesthesia and conventional treatment supporting regional anesthesia techniques [33]. Moreover, Kairaluoma et al. reported a reduction of chronic pain intensity by preemptive PVB in breast surgery patients [34]. In this study, a single preemptive injection of local anesthetic was performed.

Our study has some limitations. Although statistical significance was reached, the number of participants was limited. Ketamine was administered as a single dose only and this might be a reason for ketamine’s lack of effectiveness on acute pain intensity. Only a single anesthesiologist performed each PVB in our study. Pain intensity was evaluated at rest but not during coughing. The lower pain severity and decreased morphine consumption in the PVB group could be also associated with fentanyl added to the local anesthetic solution. Thus, patients in the PVB group also received from 240 to 384 mcg of fentanyl which is approximately 18 to 29 mg of morphine. Pulmonary function tests were not performed in this study. Only ward nurses were blinded in our study. 

To conclude, PVB as a pain treatment strategy reduces acute and chronic pain intensity in patients after thoracotomy. A single dose of ketamine does not decrease pain intensity after thoracic surgery.

## Figures and Tables

**Figure 1 jcm-09-00793-f001:**
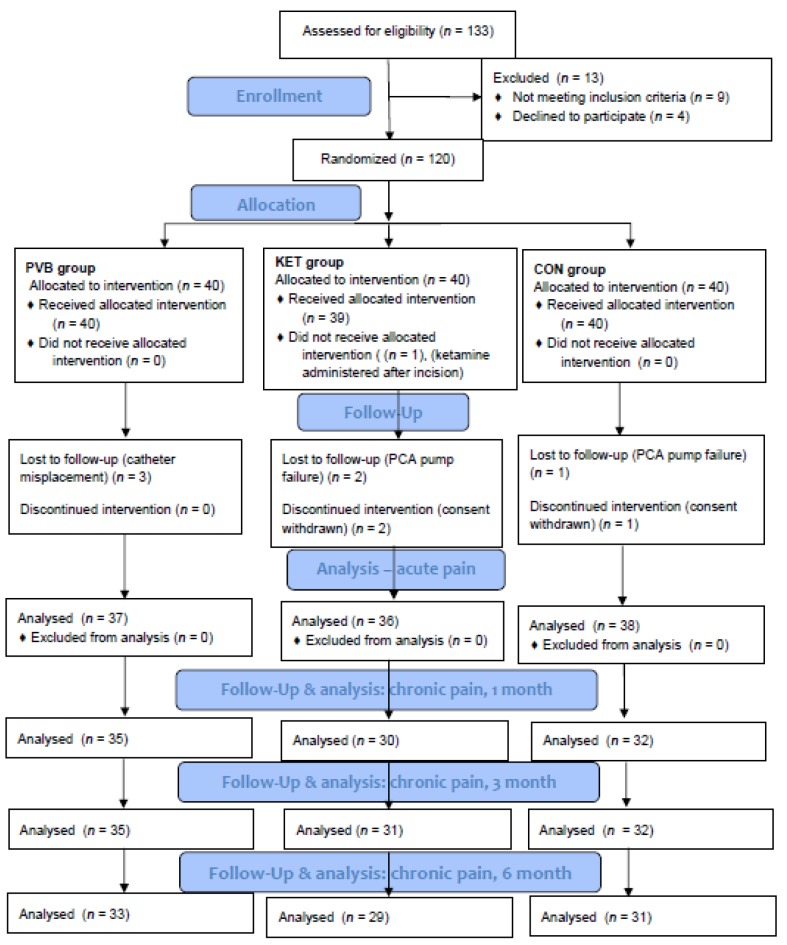
Study flowchart. CON denotes control group, KET denotes ketamine group, PVB denotes paravertebral block group, PCA denotes patient-controlled analgesia.

**Table 1 jcm-09-00793-t001:** Patient demographics.

Features	PVB (*n* = 37)	KET (*n* = 36)	CON (*n* = 38)	*p* Value
Age, years	62.3 (59.3–65.2)	56.4 (52.2–60.5)	58.2 (54.7–61.8)	0.06
Weight, kg	71.7 (68.6–74.8)	72.0 (68.0–76.0)	67.4 (64.2–70.6)	0.1
Height, cm	166.4 (163.3–169.6)	169.0 (166.2–171.8)	165.9 (163.0–168.8)	0.27
BMI, kg/m^2^	26.0 (24.9–27.1)	25.2 (24.0–26.4)	25.0 (23.4–25.8)	0.25
Women	14 (38)	9 (25)	16 (42)	0.53
Surgery time, min	106.1 (90.4–121.8)	105.0 (86.3–123.7)	107.4 (92.2–122.5)	0.98

Results are presented as means (95% confidence intervals) or *n* (%). *n* denotes sample size, BMI denotes Body-mass index, CON denotes control group, KET denotes ketamine group, PVB denotes paravertebral block group.

**Table 2 jcm-09-00793-t002:** The table presents subsequent results of visual analog scale intensity in mm after lateral thoracotomy.

Hours of Post-Surgery	PVB	KET	CON	*p* Value
0 h	38 (21–61) **	64.5 (51–84)	72 (54–84)	<0.001
2 h	29 (19–42) **	45 (41–53.5)	55 (42–63)	<0.001
4 h	26 (21–40) *	35 (22.5–50)	40 (28–52)	0.014
8 h	25 (21–35)	30.5 (19–40.5)	32.5 (21–40)	0.28
12 h	20 (11–29)	23 (11–35.5)	25 (14–33)	0.68
24 h	16 (10–23) **	21.5 (14–31)	27.5 (19–35)	0.001

Data are shown as medians (interquartile ranges). Probability was calculated with the Kruskal–Wallis test by ranks. If this test showed a significant result, a pairwise comparison was made with the Mann–Whitney U test. A significant calculated probability was set at 0.017 after Bonferroni correction. * CON is significantly higher than PVB; ** CON and KET are significantly higher than PVB. CON denotes controlled group, KET denotes ketamine group, PVB denotes paravertebral block group.

**Table 3 jcm-09-00793-t003:** The severity of post-thoracotomy pain syndrome detected with NPSI (0–10) as medians (interquartile ranges).

After Discharge	PVB	KET	CON	*p* Value
1 month	17.5 (10–21) *	24 (14–39)	21.5 (16.5–29.5)	<0.001
3 months	6 (2–8) *	16 (9–24)	9.5 (5–17)	<0.001
6 months	3.5 (0–8) *	11.5 (8–18)	10 (4–21)	<0.001

Probability was calculated with the Kruskal–Wallis test by ranks. If this test showed a significant result, a pairwise comparison was made with the Mann–Whitney U test. A significant calculated probability was set at 0.017 after Bonferroni correction. * CON and KET are significantly higher than PVB. CON denotes the control group, KET denotes ketamine group, PVB denotes paravertebral block group. NPSI denotes Neuropathic Pain Syndrome Inventory.

**Table 4 jcm-09-00793-t004:** The table presents the number of patients who perceived chronic postsurgical pain at 1, 3, and 6 months after thoracotomy.

Time after Surgery	Number of Patients (%)			Probability
	PVB	KET	CON	
1 month	32 (91)	30 (100)	32 (100)	0.1
3 month	29 (83)	31(97)	30 (97)	0.1
6 month	16 (48)	26 (90)	28 (90)	<0.001

Probability was calculated with the Freeman-Halton extension of Fisher’s exact test. CON denotes the control group, KET denotes ketamine group, PVB denotes paravertebral block group.

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
