# Peer review of "Paravertebral Block Versus Preemptive Ketamine Effect on Pain Intensity after Posterolateral Thoracotomies: A Randomized Controlled Trial"

_jcm, 2020, doi:10.3390/jcm9030793_

Round 1
Reviewer 1 Report
In their manuscript, entitled “Paravertebral block versus preemptive ketamine effect on pain intensity after posterolateral thoracotomies: a randomized controlled trial”, the authors are presenting the results of a clinical study comparing a regional anesthesia technique with preemptive intravenous ketamine regarding the incidence of severe acute and chronic pain after thoracic surgery.
The overall idea of the study is interesting and the manuscript is well-written. However, there some issues within this manuscript, which I would kindly like to ask the authors to address:
- Abstract: I would recommend the authors to already mention in the abstract that they have used a catheter technique for the PVB. In the current version of the manuscript this issue is not clear in the beginning and might mislead the readers to think of the PVB as a single-shot technique.
- Methods: It seems that the PVB patients also received a dose of fentanyl via the PVB catheter. However, to me it is not obvious, whether there was a continuous application of fentanyl intra- and – most importantly – postoperatively as well. If fentanyl had still been administered postoperatively, I would consider this as a major shortcoming of the study, as the comparison between the groups regarding morphine consumption might be difficult. Therefore, this issue should be clarified.
- Methods: A single dose of 2g of metamizol as reported in the methods section is double the amount of the drug recommended by the manufacturers for a single dose. What was the rationale for such a high dose?
Author Response
Reviewer 1.
Sir, thank you for your comments. We believe that they improved our manuscript.
- Abstract: I would recommend the authors to already mention in the abstract that they have used a catheter technique for the PVB. In the current version of the manuscript this issue is not clear in the beginning and might mislead the readers to think of the PVB as a single-shot technique.
Our response
In line 13 of the Abstract, the word “continuous” was added to highlight that it was not a single-shot PVB.
- Methods: It seems that the PVB patients also received a dose of fentanyl via the PVB catheter. However, to me it is not obvious, whether there was a continuous application of fentanyl intra- and – most importantly – postoperatively as well. If fentanyl had still been administered postoperatively, I would consider this as a major shortcoming of the study, as the comparison between the groups regarding morphine consumption might be difficult. Therefore, this issue should be clarified.
Our response
Thank you for this criticism. The local anesthetic mixture has been used in the hospital for years. We did not want to change it to avoid problems with staff members who prepared the solution but were not directly involved in the study. However, after the Reviewer remark, we calculated the morphine equivalent of fentanyl, it appeared that it was approximately 18 to 29 mg of morphine per day. We added this information to the Limitation section to present readers that it could have had a significant impact on our study results.
- Methods: A single dose of 2g of metamizol as reported in the methods section is double the amount of the drug recommended by the manufacturers for a single dose. What was the rationale for such a high dose?
Our response
Interestingly, according to the European Medicines Agency (EMA), “the maximum daily dose of metamizole reflected in the Product Information of the different medicinal products varies from 1.5 g to 6 g”. Due to this disinformation, now, EMA recommends 4 g per day of metamizole, however, “it seems appropriate to allow, if necessary, a parenteral single dose of 2500 mg metamizole and a maximum daily dose of 5000 mg metamizole”. In our study, this maximum daily dose was used because of suspected severe postoperative pain. Here, we attached the link to this document - 2018 EMA/143912/2019 (https://www.ema.europa.eu/en/documents/referral/metamizole-article-31-referral-chmp-assessment-report_en.pdf)
Reviewer 2 Report
I read the manuscript entitled "Paravertebral block versus preemptive ketamine effect on pain intensity after posterolateral thoracotomies: a randomized controlled trial" submitted.
The authors claim that they demonstrate that paravertebral block both acute and chronic pain after thoracotomies.
The study was well-organized, preformed fairly, and analyzed properly.
Just minor comments
Table 2
"Hous of surgery" is misleading
Hous of post-surgery can be used.
Table 4
"1 month" is written in bold font and underlined.
Limitation
Only one anesthesiologist performed paravertebral block. I think this is a limitation of this study. The authors should mention this limitation.
Author Response
Sir, thank you for your remarks
- Table 2 "Hous of surgery" is misleading Hous of post-surgery can be used.
Our response
According to the Reviewer suggestion an expression “hours of surgery” was modified to” hours of post-surgery”
- Table 4 "1 month" is written in bold font and underlined.
Our response
The expression “1 month” is not bolded and underlined in Table 4.
- Limitation Only one anesthesiologist performed paravertebral block. I think this is a limitation of this study. The authors should mention this limitation.
Our response
We added the information regarding performing PVB by a single anesthesiologist to the Limitations.
Reviewer 3 Report
The article appears well written and the study methodology well performed.
this is an interesting article. In my opinion, it does not require revisions.
Author Response
Sir, thank you for your evaluation.